# Theoretical Analysis of the Built-in Metabolic Pathway Effect on the Metabolism of Erythrocyte-Bioreactors That Neutralize Ammonium

**DOI:** 10.3390/metabo11010036

**Published:** 2021-01-06

**Authors:** Evgeniy Protasov, Larisa Koleva, Elizaveta Bovt, Fazoil I. Ataullakhanov, Elena Sinauridze

**Affiliations:** 1Laboratory of Physiology and Biophysics of the Cell, Center for Theoretical Problems of Physicochemical Pharmacology, Russian Academy of Sciences, Srednyaya Kalitnikovskaya Str., 30, Moscow 109029, Russia; ie.bovt.rv@gmail.com (E.B.); ataullakhanov.fazly@gmail.com (F.I.A.); 2Laboratory of Biophysics, Dmitriy Rogachev National Medical Research Center of Pediatric Hematology, Oncology, and Immunology, Ministry of Healthcare, Samory Mashela Str., 1, GSP-7, Moscow 117198, Russia; 3Department of Physics, Lomonosov Moscow State University, Leninskie Gory, 1, Build. 2, GSP-1, Moscow 119991, Russia; 4Department of Molecular and Translational Medicine, Moscow Institute of Physics and Technology, Institutskiy per., 9, Dolgoprudny 141701, Russia

**Keywords:** ammonium removal, erythrocyte, erythrocyte-bioreactor, limitations of the erythrocyte-bioreactor efficiency, mathematical modeling

## Abstract

The limitations of the efficiency of ammonium-neutralizing erythrocyte-bioreactors based on glutamate dehydrogenase and alanine aminotransferase reactions were analyzed using a mathematical model. At low pyruvate concentrations in the external medium (below about 0.3 mM), the main limiting factor is the rate of pyruvate influx into the erythrocyte from the outside, and at higher concentrations, it is the disappearance of a steady state in glycolysis if the rate of ammonium processing is higher than the critical value (about 12 mM/h). This rate corresponds to different values of glutamate dehydrogenase activity at different concentrations of pyruvate in plasma. Oxidation of reduced nicotinamide adenine dinucleotide phosphate (NADPH) by glutamate dehydrogenase decreases the fraction of NADPH in the constant pool of nicotinamide adenine dinucleotide phosphates (NADP + NADPH). This, in turn, activates the pentose phosphate pathway, where NADP reduces to NADPH. Due to the increase in flux through the pentose phosphate pathway, stabilization of the ATP concentration becomes impossible; its value increases until almost the entire pool of adenylates transforms into the ATP form. As the pool of adenylates is constant, the ADP concentration decreases dramatically. This slows the pyruvate kinase reaction, leading to the disappearance of the steady state in glycolysis.

## 1. Introduction

Enzymes can be used to treat a number of diseases. To alleviate the patient’s condition, it may be necessary to regulate the concentration of certain substances in the blood. This can be achieved by introducing enzymes into the bloodstream that catalyze reactions through which the target substance is produced or consumed. However, this method has several disadvantages. Immediately after administration, the plasma activity of the administered enzyme decreases due to the work of plasma proteases and the patient’s immune system. For most enzymes, the half-life is hours to a maximum of tens of hours [1,2,3]. In addition, the administration of a foreign protein into the body can lead to severe allergic reactions. These problems can be avoided by placing enzymes in some kind of a container where they are not available to the immune system. The patient’s own erythrocytes are well suited for the role of such container cells. Erythrocytes containing artificially-embedded metabolic pathways capable of working inside the cell, but absent in a normal erythrocyte, are called erythrocyte-bioreactors (EBRs) [4,5]. Various scientific groups have created EBRs at different times to remove ethanol [6,7], methanol [8,9], ammonium [10,11,12], asparagine [13,14,15], and other compounds from the bloodstream [16,17,18].

To assess the efficiency of EBR, it is necessary to consider factors that may limit the maximum rate of production or consumption of the target substance. There are two classes of such factors. The first is associated with the low permeability of the erythrocyte membrane for substrates or products of the incorporated metabolic pathway. With a low membrane permeability for the substrate, the rate of the target reaction is limited by the rate of entry of this substrate into the cell from the blood plasma. In the case of low membrane permeability for the reaction product or accumulation of glycolysis metabolites upon loss of the glycolysis steady state, these products accumulate in the cell, which can lead to its death as a result of osmotic lysis. The second class of limitations is associated with the interaction of the built-in metabolic pathway with the erythrocyte’s own metabolism. These interactions can lead to a variety of consequences from decreased EBR efficiency to cell death. The purpose of this work was to study the described limitations of using erythrocyte-bioreactors to remove ammonium (ammocytes). These EBRs may be useful for the treatment of hyperammonemia, a pathological condition caused by an excess of ammonium in the patient’s blood plasma [19]. Ammonium in high concentrations is toxic to the central nervous system. Hyperammonemia can be caused by both congenital defects in the enzymes of the urea cycle and chronic liver diseases or acute intestinal infections. Current methods of decreasing ammonium concentration are not satisfactorily efficient. It usually takes 2 to 10 days for a patient’s blood ammonium concentration to decrease to normal levels (≤60 μM) [12]. Thus, the problem of effective treatment of hyperammonemia remains.

Various scientific groups have created EBRs for ammonium removing based on two different enzymes: glutamine synthetase (GS) or glutamate dehydrogenase (GDH). GS catalyzes, in the presence of adenosine triphosphate (ATP), the formation of glutamine (GLN) and adenosine diphosphate (ADP) from ammonium and glutamate (GLU) (reaction (1)) [11,20]. GDH, in the presence of NADPH, catalyzes the formation of glutamate from ammonium and α-ketoglutarate (AKG) (reaction (2)) [10,21,22]:GS:   GLU + NH_4_^+^ + ATP ↔ GLN + ADP + PO_4_^−3^,(1)
GDH: AKG + NH_4_^+^ + H^+^ + NADPH ↔ GLU + NADP + H_2_O(2)

However, in vivo experiments have shown that, although the content of FITC-labeled ammocytes loaded with GDH in the bloodstream does not practically decrease within 48 h, such EBRs can remove ammonium from the bloodstream of mice only during a short period of time (0.5–1 h) [10,11]. A theoretical study [12] showed that the reason for this is the depletion of the substrate (glutamate or α-ketoglutarate, respectively) inside the cell, which is due to the low permeability of the erythrocyte membrane for these substances. In the same work, a new variant of the metabolic pathway for ammonium consumption was proposed, consisting of two sequential reactions catalyzed by glutamate dehydrogenase (2) and alanine aminotransferase (AAT), which converts glutamate in the presence of pyruvate (PYR) into AKG and alanine (ALA) (reaction (3)):AAT: GLU + PYR ↔ AKG + ALA.(3)

As a result, the problem of the depletion of substrates (AKG and GLU) in the EBRs was solved, as they were consumed and produced in a cyclic process; as a result, the rate of ammonium consumption did not depend on their influx into erythrocytes from plasma. The rate of ammonium consumption by such EBRs in vivo in the model of hyperammonemia in mice was (under the selected experimental conditions) 2 mmol/h × L_EBRs_. It is assumed that this rate can be increased by increasing the activity of the encapsulated enzymes in the cell. However, since GDH uses NADPH in its work, which is simultaneously used by other processes in glycolysis, it is difficult to determine in advance how high concentrations of the built-in enzyme can affect cell metabolism. The aim of this work was to analyze such limitations as well as limitations associated with the rate of penetration of the reaction substrate into the cell using a mathematical model of ammocytes.

## 2. Results and Discussion

We showed earlier [12] that the efficiency of EBRs containing GDH and AAT is not associated with the transport of AKG and/or GLU across the cell membrane. Under these conditions, the limitation on the rate of ammonium consumption in the considered EBRs is associated with the transport of pyruvate through the erythrocyte membrane and with a change in the proportion of NADPH in the total pool (NADP + NADPH) during the work of the built-in enzyme system in the erythrocyte.

When the glycolysis system and built-in enzymes are in a stationary state, the right parts of all ordinary differential equations (ODEs) describing this system are equal to 0. The rate of glycolysis should be equal to the rates of pyruvate kinase or LDH reactions:*V_glycolysis_* = *V_PK_* = *V_LDH_*,(4)
where *V* denotes the rate of the corresponding reaction.

Thus, we have:*d*[*PYR*]/*dt* = *V_PK_* − *V_LDH_* − *V_AAT_* + *V_trPYR_* = 0,(5)
*d*[*GLU*]/*dt* = *V_GDH_* − *V_AAT_* = 0(6)
where *V_trPYR_* is the rate of pyruvate flow from plasma into the erythrocyte, which depends on the concentration of pyruvate in plasma.

Considering Equations (4)–(6), we obtain:*V_GDH_* = *V_trPYR_*(7)

Thus, the maximum rate of ammonium consumption in the system in a steady state is equal to the rate of pyruvate influx into the cell from the outside. The higher the plasma pyruvate concentration, the higher the rate.

The dependence of the maximum rate of pyruvate influx into the erythrocyte (which is equal to the maximum rate of ammonium consumption by the EBR) on the plasma pyruvate concentration is presented in Figure 1A. The plasma concentrations of all other metabolites are equal to the physiological ones. For example, at a physiological plasma concentration of pyruvate (about 0.07 mM), the maximum possible rate of its influx into the cell is about 3.5 mM/h.

Apart from the plasma concentration of pyruvate, the efficiency of the ammonium-neutralizing system also depends on the activity of the GDH encapsulated into the erythrocyte. At low external pyruvate concentrations (less than 0.3 mM), there is no encapsulated GDH activity, which can provide an ammonium consumption rate sufficient to overcome the steady state in glycolysis. At higher external pyruvate concentrations, there is a critical value of GDH activity (which is significantly lower than the maximum pyruvate influx), above which the steady state in glycolysis is lost. This critical activity differs at different external pyruvate concentrations, but always corresponds to an ammonium consumption rate of about 12 mM/h (red dashed line in Figure 1A). When the critical activity of GDH is exceeded, the steady state in glycolysis disappears and all glycolytic metabolites begin to accumulate, except for glucose-6-phosphate (G6P), fructose-6-phosphate (F6P), nicotinamide adenine dinucleotide (NAD), and nicotinamide adenine dinucleotide phosphate (NADP) the concentrations of which, although changing, gradually reach a new steady state.

Figure 1B presents the dependence of the critical activity of GDH, which can be incorporated into the erythrocyte without loss of the steady state in glycolysis, on the extracellular concentration of pyruvate. This curve has a vertical asymptote at low external pyruvate concentrations. This means that at low (less than 0.3 mM) external pyruvate concentrations, the critical value of the ammonium consumption rate cannot be achieved with any encapsulated GDH activity since the maximum rate of ammonium neutralization in this area is limited by the pyruvate influx into the cell. This can be observed in the physiological state. At higher pyruvate concentrations, the maximum possible activity of encapsulated GDH is limited by a decrease in the NADPH fraction in the pool (NADP + NADPH), which leads to the disappearance of the steady state in glycolysis.

Figure 2 demonstrates the change in the erythrocyte concentrations of some metabolites at a plasma pyruvate concentration of 1 mM and different activities of the incorporated GDH. We conducted a detailed study of the reasons for the disappearance of the steady state in glycolysis in these experiments.

As mentioned above, when the critical value of the encapsulated GDH activity is exceeded, the steady state in glycolysis disappears. Since cell lysis occurs with an increase in the total intracellular concentration of metabolites by tens of mM compared to the normal physiological concentration, such dynamics leads to the osmotic lysis of the erythrocyte within several hundred hours [12,23].

Since glycolysis and the system of enzymes built into the erythrocytes overlap at the NADPH level, the key parameter influencing the loss of the stationary state in glycolysis is the NADP/(NADP + NADPH) ratio, which, in the normal physiological state, is maintained near zero due to the reduction of NADP to NADPH in reactions of the pentose phosphate pathway. Oxidation of NADPH in a normal erythrocyte occurs in the glutathione reductase reaction, and its rate is determined by the rate of glutathione oxidation (i.e., in total, by the rate of oxidative processes in the cell). The GDH reaction also oxidizes NADPH, placing additional stress on the pentose phosphate pathway. As a result, the NADP/(NADP + NADPH) ratio increases, resulting in the activation of the pentose phosphate pathway (PPP), which is associated with glycolysis via common metabolites (Figure 3). Earlier [24], when a joint mathematical model of glycolysis and PPP was investigated, at high fluxes in the PPP, glycolysis regulation systems were found to be unable to stabilize the level of ATP in the erythrocyte, which begins to grow and reaches a value close to the total pool of adenylates (Figure 2A). At the same time, the concentration of ADP sharply decreases, resulting in the flow of metabolites, formed in the upper part of glycolysis, being unable to completely pass through the pyruvate kinase reaction (Figure 3). As a result, their accumulation begins and the steady state in glycolysis disappears.

The kinetics of ATP, NADP, 2,3-DPG, and the total concentration of other glycolytic metabolites (from G6P to PEP excluding 2,3-DPG) are shown in Figure 2. These kinetics are different at different activities of encapsulated GDH. At sufficiently low activities of the incorporated GDH (for example, 10 IU/mL_EBRs_, curves 2), the changes in these concentrations in comparison with the corresponding stationary concentrations of these metabolites in the erythrocytes without encapsulated enzymes (curves 1) are insignificant. The concentrations of all metabolites under these conditions reach a new steady state. If the activity of GDH encapsulated in the erythrocyte is higher than a certain threshold (for example, 30 or 60 IU/mL_EBR_s, curves 3 and 4), all metabolites shown in Figure 2 begin to accumulate without limit, indicating the loss of the steady state in glycolysis.

To determine the threshold value of GDH activity in erythrocytes above which there is a loss of the steady state in glycolysis, the dependences of the total concentration of all glycolytic metabolites in the erythrocyte on the NADP/(NADP + NADPH) ratio were calculated for different activities of the encapsulated GDH, as well as in the erythrocyte without encapsulated enzymes, but at different established fixed stationary levels of NADP (Figure 4). Loss of steady state was determined as a significant increase in the total concentration of all glycolysis metabolites above the physiological level of this parameter. Until the value of the NADP/(NADP + NADPH) ratio is about 0.35–0.40, the total concentration of all glycolysis metabolites remains constant and approximately equal to the initial value (Figure 4). Above this value, accumulation of glycolysis metabolites is observed, which indicates the loss of the steady state in the system. These dependences behave similarly for the system with built-in enzymes that consume ammonium (curve 1) and for the system without built-in metabolic pathways, but with different established fixed NADP values (provided at *d*[*NADP*]/*dt* = 0, curve 2).

The value of the NADP/(NADP + NADPH) ratio can change significantly with an increase in the oxidative load in the PPP. Figure 4B shows that with an increase in the rate of oxidative processes in the cell (*Vox*), the critical value of the GDH activity into the cells decreases (shifts to the left), which does not cause the loss of the steady state in glycolysis. It illustrates that oxidative processes promote NADPH oxidation and, therefore, an increase in PPP flux, whose main function is to maintain the NADPH concentration. However, Figure 4B also shows that the system is not very sensitive to increases in oxidative stress, since an increase in the rate of oxidative processes by a factor of 10 leads to a decrease in the critical activity of GDH by only about 10% (curves 1 and 2 in Figure 4B). This shows that such bioreactors are able to operate over a wide range of oxidative loads.

Calculations showed that upon loss of the steady state, all glycolysis metabolites accumulate up to PEP, inclusively. PEP is a substrate for pyruvate kinase (PK), which, in the presence of ADP, catalyzes the irreversible conversion of PEP to PYR and ATP. The rate of this reaction depends on the state of the adenylate pool. We then investigated this dependence on a simplified system that included only one PEP transformation reaction.

Since the rate of PEP production is determined by the glycolysis reactions preceding pyruvate kinase, to simplify the system, we considered it constant and equal to *V_PEP_*. The rate of PEP consumption is equal to the rate of the pyruvate kinase reaction. The resulting system will be described by Equations (8)–(10):*d*[*PEP*]/*dt* = *V_PEP_* − *V_PK_*,(8)
*V_PEP_* = *const*,(9)
(10)$$V_{PK}=A_{PK}\frac{\frac{\left[PEP\right]\left[ADP\right]}{K_{PK}^{1}K_{PK}^{2}}}{1+\frac{\left[ATP\right]}{K_{PK}^{3}}+\frac{\left[ADP\right]}{K_{PK}^{2}}+\frac{\left[PEP\right]}{K_{PK}^{1}}+\frac{\left[PEP\right]\left[ADP\right]}{K_{PK}^{1}K_{PK}^{2}}}\;$$
where *A_PK_* = 120 mM/h, *K*^1^*_PK_* = 0.05 mM, *K*^2^*_PK_* = 0.43 mM, and *K*^3^*_PK_* = 0.35 mM [12]. Reaction rates are given in mmol/h × L_RBCs_.

Since, in the vicinity of the state of interest to us, more than 95% of the adenylate pool (*P*_0_ = *ATP + ADP + AMP*) is in the form of ATP, we take the AMP concentration as equal to 0. To simplify further calculations, we also introduce the following notation: $a_{1}=\frac{1}{K_{PK}^{1}}$, $a_{2}=\frac{1}{K_{PK}^{2}}$, $a_{3}=\frac{1}{K_{PK}^{3}}$, $a_{4}=\frac{1}{K_{PK}^{1}K_{PK}^{2}}$; $P_{0}=ATP+ADP$ (AMP=0), $\alpha =\frac{ATP}{ATP+ADP}$, $ATP=\alpha P_{0}$, $ADP=\left(1-\alpha \right)P_{0}$, PEP=x.

Then, the singular point (*x*_0_) corresponding to a stationary state can be determined from Equations (11)–(13):*V_PEP_* − *V_PK_* = 0,(11)
where:(12)$$V_{PEP}=\frac{A_{PK}a_{4}\left(1-\alpha \right)P_{0}x_{0}}{1+a_{3}\alpha P_{0}+a_{2}\left(1-\alpha \right)P_{0}+a_{1}x_{0}+a_{4}\left(1-\alpha \right)P_{0}x_{0}}\;,$$
(13)$$x_{0}=\frac{V_{PEP}+a_{3}V_{PEP}\alpha P_{0}+a_{2}V_{PEP}\left(1-\alpha \right)P_{0}}{a_{4}P_{0}\left(A_{PK}-V_{PEP}\right)\left(1-\alpha \right)-a_{1}V_{PEP}}\;$$

Our analysis showed that the singular point remains stable everywhere. From the dependence of the singular point position on parameter *α* (Figure 5), we found that this dependence has a vertical asymptote at *α* = 0.995. To the right of this point, there is no stationary value of PEP concentration (*x*_0_ < 0). This result, obtained on the reduced model, agrees well with the results obtained from the full model, where the disappearance of the steady state was also observed at values of *α* close to 0.99.

## 3. Methods

### 3.1. Erythrocyte-Bioreactors for Ammonium Removing

In this work, we considered the metabolic pathway for ammonium removing, which consists of two reactions catalyzed by glutamate dehydrogenase (2) and alanine aminotransferase (3). As a result, the AKG consumed in reaction (2) is re-formed in reaction (3). The resulting cyclic pathway avoids the problems due to the low permeability of the erythrocyte membrane for AKG [25] and accumulation of GLU, but may be dependent on pyruvate influx from external media into erythrocytes. At the same time, alanine, produced in the AAT reaction (3), passes relatively easily through the erythrocyte membrane [26]. This is an important characteristic of the product of the built-in metabolic pathway because the accumulation of high concentrations of the substance in the cell after some time should lead to its death due to osmotic lysis.

Since the incorporated metabolic pathway uses NADPH as a substrate (with NADH, this pathway under physiological conditions does not consume but produces ammonium [12]), it significantly interacts with the pentose phosphate pathway (PPP). The first oxidative phase of this pathway (including reactions of G6PDH and 6PGLDH), in the presence of NADP, directly oxidizes and then decarboxylates G6P to NADPH and a mixture of pentose phosphates. Subsequent nonoxidative stages of PPP again lead to the formation of metabolites involved in glycolysis (glucose-6-phosphate, fructose-6-phosphate, and glyceraldehyde phosphate). The entry reaction from glycolysis into PPP via G6P (via the G6PDH reaction) is irreversible (Figure 3). This is the rate-limiting stage of PPP. Thus, glycolysis and PPP should be considered together when simulating ammonium processing. A complete scheme of the metabolic system of erythrocyte glycolysis in the presence of built-in GDH and AAT was presented earlier [12]. Here, we only present a simplified diagram showing the relationship between glycolysis, PPP, and embedded reactions (Figure 3). Although all reactions of the system, except for hexokinase (HK), G6PDH, phosphofructokinase (PFK), and pyruvate kinase (PK) reactions, are reversible, for simplicity, only arrows are shown in the diagram, indicating the main direction of the reactions under normal conditions. In addition, NAD/NADH are excluded from the scheme.

### 3.2. Mathematical Model of EBR Neutralizing Ammonium

We previously developed and described the mathematical model analyzed in this work [12]. This is based on the model of glycolysis in human erythrocytes described by Martinov et al. [23].

We added equations to this model that describe the transport of pyruvate and lactate across the erythrocyte membrane in more detail, the reactions of the ammonium neutralizing system, and the G6PDH reaction. The model contains 19 ordinary differential equations (ODEs). The complete system of equations, kinetic constants, and expressions for the rates of enzymatic reactions were previously described [12]. The stationary concentrations of glycolysis metabolites in normal erythrocytes were used as the initial conditions.

In the previous article [12], the model was verified in vitro (and in vivo in mice). Good agreement was obtained between experimental results and model simulations.

The assumptions of the model are briefly summarized here, since a full description of these assumptions was previously provided [12]:
The rate of the hexokinase reaction was considered independent of the glucose concentration;The erythrocyte membrane was considered impermeable to all the substances under consideration, except for lactate, pyruvate, α-ketoglutarate, alanine, and ammonium ions;The rate of transport of all metabolites penetrating the membrane, except for lactate and pyruvate, was considered proportional to the gradient of their concentrations on both sides of the membrane:
*V_trA_* = *K_A_* × ([*A*]*_int_* − [*A*]*_ext_*),(14)
where *K_A_* is the permeability coefficient for metabolite *A*, and [*A*]*_int_* and [*A*]*_ext_* are the concentration of *A* in the cell and in the plasma, respectively.

The rate of transport of lactate and pyruvate is described by equations obtained on the basis of the data [27]:(15)$$V_{trPYR}=A_{trPYR}\frac{{\left[PYR\right]}_{ext}-\left[PYR\right]}{{\left[PYR\right]}_{ext}+\left[PYR\right]+\left(1+\frac{\left[LAC\right]+{\left[LAC\right]}_{ext}}{K_{I}^{trPYR}}\right)K_{m}^{trPYR}\;}\;,$$
where $A_{trPYR}=120\;\mathrm{mM}/h$; $K_{m}^{trPYR}=1.9\;\mathrm{mM}$; $K_{I}^{trPYR}=11\;\mathrm{mM}$; and
(16)$$V_{trLAC}=A_{trLAC}\frac{{\left[LAC\right]}_{ext}-\left[LAC\right]}{{\left[LAC\right]}_{ext}+\left[LAC\right]+\left(1+\frac{\left[PYR\right]+{\left[PYR\right]}_{ext}}{K_{I}^{trLAC}}\right)K_{m}^{trLAC}}\;,$$
where $A_{trLAC}=120\;\mathrm{mM}/h$; $K_{m}^{trLAC}=9\;\mathrm{mM}$; $K_{I}^{trLAC}=1.6\;\mathrm{mM}$.

Concentrations with an *ext* index refer to the extracellular medium, and concentrations without an index refer to the intracellular medium. All plasma concentrations were considered constant.
4.The transport of ammonium was considered fast enough that its concentrations in plasma and cells remained in equilibrium at any time;5.Pools of adenylates, erythrocyte volume, and rate of NADPH oxidation in oxidative metabolism were considered constant;6.The pentose phosphate pathway is represented by a single G6PDH reaction, since it is the rate-limiting reaction in the PPP.

For all data on intra- and extracellular concentrations of metabolites passing through the membrane of erythrocytes, as well as the corresponding permeability coefficients, see [12]. In all calculations, the ratio of the encapsulated GDH:AAT activities was 1:5 because, as was shown earlier [12], if the AAT activity is at least five times higher than the GDH activity, AAT is no longer the limiting stage of the process. Numeric solutions of the ODE system were obtained by Runge–Kutta methods in MATLAB.

## 4. Conclusions

In this work, only metabolic causes are considered that can lead to the rapid death of erythrocyte bioreactors in the bloodstream. Although the loading of enzymes into cells in some cases can cause changes in other properties of erythrocytes, such as biodegradability, biocompatibility, safety, and biodistribution of carrier-erythrocytes, these effects for each type of bioreactor should be studied in experimental work. Therefore, we believe that we can restrict ourselves to indicating that the lifetime of carrier-erythrocytes obtained by soft dialysis methods (discussed in this work) in the bloodstream is 30 days or more (see, for example, [14]).

The efficacy of erythrocyte-bioreactors removing ammonium from the blood, incorporating a metabolic pathway from two consecutive reactions (GDH and AAT), one of which consumes NADPH during operation, is limited by two factors: the rate of pyruvate influx in the cell (at low pyruvate concentrations in the medium) and the interaction of GDH with the pentose phosphate pathway through the oxidation of NADPH (at higher concentrations of pyruvate in the medium). In the second case, with an increase in the activity of GDH inside the cell, the steady state in glycolysis disappears and intermediate metabolites begin to accumulate, ultimately leading to osmotic lysis of the cell. Thus, to preserve the viability of EBRs, at high concentrations of pyruvate in the extracellular medium, the GDH activity in the cell should not exceed the critical value. This value is different at different concentrations of pyruvate in the external medium and corresponds to the rate of the glutamate dehydrogenase reaction of about 12 mM/h.

With increasing GDH activity, the flux through PPP increases due to the oxidation of NADPH and an increase in the stationary portion of NADP in the total pool (NADP + NADPH). In this case, glycolysis regulatory systems become unable to stabilize the level of ATP in the cell, which increases and begins to account for more than 95% of the total pool of adenylates [24]. With a high (above 99%) portion of ATP in the pool of adenylates, the rate of pyruvate kinase reaction decreases so much due to a decrease in the concentration of ADP that it becomes unable to consume PEP at the rate at which PEP is produced in the previous stages of glycolysis. This is the direct reason for the loss of the steady state.

The limitations described in this work are typical not only for ammonium-neutralizing EBRs but also for any EBRs that use glycolysis metabolites or other natural erythrocyte metabolic pathways. Since the external concentration of pyruvate cannot be significantly changed in the patient’s blood, the disappearance of the steady state of glycolysis in vivo for the studied ammocytes should not be expected. However, similar phenomena may be observed in other EBRs, the efficiency of which does not depend on the influx of pyruvate into cells.

Since the physiological concentration of pyruvate in erythrocytes is low (about 0.1 mmol/L_RBCs_), the rate of ammonium removing by the studied EBRs under these conditions is determined by the rate of pyruvate influx into erythrocytes from plasma. For physiological concentrations of pyruvate, this rate is about 4 mmol/h × L_RBCs_. These data enable the estimations of the rate of ammonium removal from plasma, provided that a person with a blood volume of 5 L is injected with 200 mL of such EBRs (in recalculation to hematocrit 100%). For the physiological state, the amount of consumed ammonium, which can be provided by this volume of EBRs is 0.2 L_EBRs_ × 4 mmol/L_EBRs_ × h = 0.8 mmol/h (per 5 L of blood) or 19.2 mmol of ammonium (per day per 5 L of blood). As a result, the concentration of ammonium in the blood should decrease by 19.2 mmol/5 L_blood_ = 3.84 mmol/L_blood_ (per day). The calculated rate is the rate of ammonium consumption by the administered EBRs, but the actual steady-state plasma ammonium concentration in the hyperammonemic patients also depends on the rate of ammonium production in their body. Data on the level of ammonium production in patients with hyperammonemia are very scarce, or completely lacking. However, the obtained rate of ammonium consumption is high enough to hope that the proposed EBRs will be able to effectively reduce the steady-state concentration of ammonium in a patient’s body for a long time.

## Figures and Tables

**Figure 1 metabolites-11-00036-f001:**
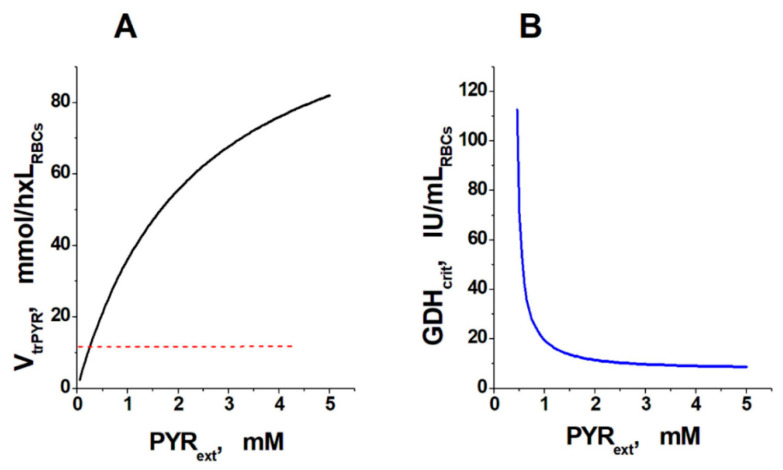
Effect of plasma pyruvate concentration (PYR_ext_) on the rate of pyruvate influx into the cell (V_trPYR_) and the critical activity of GDH that can be incorporated into erythrocytes (GDH_crit_). (**A**) Dependence of the pyruvate influx rate into erythrocytes, which corresponds to the maximum possible ammonium consumption rate, on the pyruvate concentration in plasma. The concentrations of all other metabolites are physiological. The critical ammonium consumption rate (values exceeding this lead to the disappearance of the steady state in glycolysis) is depicted by the red dashed line. (**B**) Dependence of the critical activity of GDH (corresponding to the critical rate of ammonium consumption, represented by the dashed line in panel A), which can be encapsulated in erythrocytes without loss of the steady state in glycolysis, on the concentration of pyruvate in the medium.

**Figure 2 metabolites-11-00036-f002:**
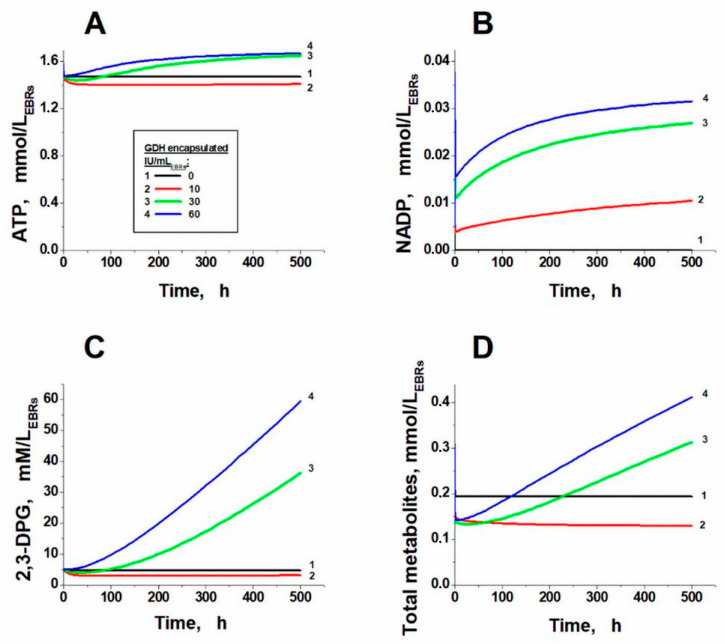
Kinetics of glycolysis metabolites in erythrocyte-bioreactors (EBRs) with different activities of encapsulated ammonium-neutralizing enzymes. Concentrations of ATP (**A**), NADP (**B**), 2,3-diphosphoglycerate (2,3-DPG) (**C**), and the sum of concentrations of glycolysis metabolites from G6P to phosphoenolpyruvate (PEP) (except 2,3-DPG) (**D**). The GDH activity was 0, 10, 30, or 60 IU/mL_EBRs_ for curves 1, 2, 3, and 4, respectively. The AAT activity was 5 times higher in each case.

**Figure 3 metabolites-11-00036-f003:**
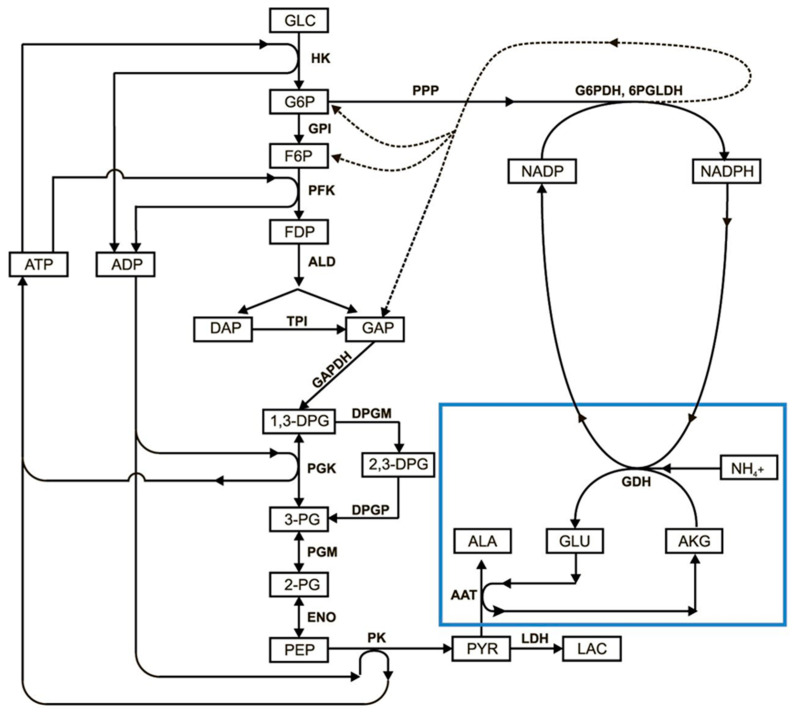
The glycolysis reactions and the reactions of encapsulated enzymes (GDH + AAT) in erythrocyte. The following abbreviations are used: GLC—glucose; G6P—glucose-6-phosphate; F6P—fructose-6-phosphate; FDP—fructose-1,6-diphosphate; DAP—dihydroxyacetone phosphate; GAP—glyceraldehyde phosphate; 1,3-DPG—1,3-diphosphoglycerate; 2,3-DPG—2,3-diphosphoglycerate; 3-PG—3-phosphoglycerate; 2-PG—2-phosphoglycerate; PEP—phosphoenolpyruvate; PYR—pyruvate; LAC—lactate; NADP—nicotinamide adenine dinucleotide phosphate; GLU—glutamic acid; AKG—α-ketoglutarate; ALA—alanine; ATP—adenosine triphosphate; ADP—adenosine diphosphate; HK—hexokinase; GPI—glucose-6-phosphate isomerase; PFK—phosphofructokinase; ALD—aldolase; TPI—triosephosphate isomerase; GAPDH—glyceraldehyde phosphate dehydrogenase; PGK—phosphoglycerate kinase; PGM—phosphoglycerate mutase; ENO—enolase; PK—pyruvate kinase; LDH—lactate dehydrogenase; G6PDH—glucose-6-phosphate dehydrogenase; 6PGLDH—6-phosphogluconate dehydrogenase; GDH—glutamate dehydrogenase; AAT—alanine aminotransferase; PPP—pentose phosphate pathway; NH_4_^+^—ammonium ion.

**Figure 4 metabolites-11-00036-f004:**
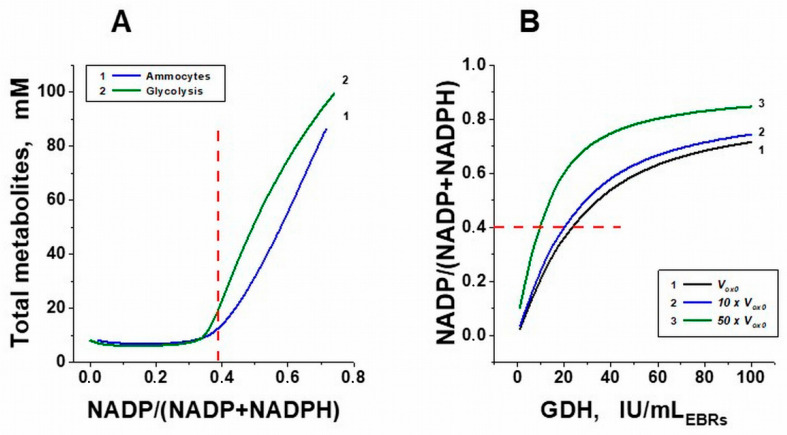
Disappearance of the steady state in glycolysis with an increase in the NADP/(NADP + NADPH) ratio. (**A**) Dependence of the total concentration of all glycolysis metabolites on the stationary value of the NADP/(NADP + NADPH) ratio 500 h after starting the system. Blue curve 1, in an erythrocyte with a built-in ammonium-consuming system; green curve 2, in an erythrocyte without an ammonium-consuming system, but at various established fixed values of stationary NADP concentrations (at *d*[*NADP*]/*dt* = 0). (**B**) Dependence of the stationary value of the NADP/(NADP + NADPH) ratio on encapsulated GDH activity at different total rates of oxidative processes in erythrocytes. The total rate of oxidative processes (*V_ox_*) corresponds to the physiological state or is increased 10 or 50 times (*V_ox_*_0_, 10 *× V_ox_*_0_, and 50 × *V_ox_*_0_, respectively). The critical value of the NADP/(NADP + NADPH) ratio above which the steady state in glycolysis disappears is noted in both panels by a red dashed line.

**Figure 5 metabolites-11-00036-f005:**
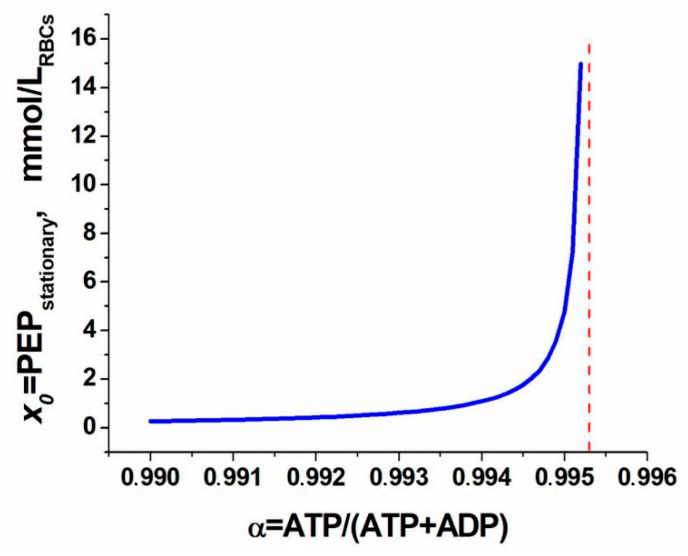
Dependence of the stationary concentration of PEP (*x*_0_) in the simplified metabolic system (Equations (8)–(10)) on the ATP fraction (*α*) in the pool of adenylates. The vertical line is the asymptote of the curve.

## Data Availability

All data are presented in this article.

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
