# Peer review of "Theoretical Analysis of the Built-in Metabolic Pathway Effect on the Metabolism of Erythrocyte-Bioreactors That Neutralize Ammonium"

_metabolites, 2021, doi:10.3390/metabo11010036_

Round 1

Reviewer 1 Report

This is an interesting article from authors who are established in the field of mathematical modeling of glycolytic pathways in erythrocyte carriers. The article is well written and I have only a few comments to make:

  • Line 52 should also address the fact that the reaction will also likely cease with metabolite accumulation 
  • It order for this work to be accessible to other workers in the field of erythrocyte carriers, the authors should clarify what they mean by 200 mL of blood. Is this packed blood? Blood haematocrit (related to cell size) has a dramatic effect on the encapsulation efficiency and efficacy of red cell carriers. The volume stated is too vague.
  • The weakness of this work is that the mathematical model has not been verified in vitro. Have the authors collected data to support this model or conducted pre-clinical studies- this should be included.

Reviewer 2 Report

12-16-20

Metabolites 1030615

RBCs are remnants of the precursor cells (reticulocytes), lacking the nucleus and organelles, circulating in the bloodstream for several months in humans. RBCs are being pursued as carriers for drug delivery for many years, with a good reason: transport of the cargoes in the circulation system is their natural vocation.

Encapsulation of cargoes into inner volume of RBC attained via transient pores in RBC membrane is being pursued by several groups. The present study aims at providing the theoretical computational and modeling guidance for this important area of biomedical research. As such, it is interesting and important.

From the standpoint of potential medical utility, however, the significance of this study depends on the pending results of characterization of the pharmacokinetics, circulation, biodistribution, biocompatibility and ultimately safety of modified RBC. This aspect, however, deserves more attention. For example, the authors acknowledge that EBR work in vivo for a very short time (lines 71-72). Based on the theoretical work they attribute this to the factors impeding enzymatic features of the bioreactor, such a insufficient accessibility of the reaction component(s). Yet, alternatively this outcome in vivo may be due to elimination of modified RBCs by spleen and liver.

In this context, this reviewer suggests to revise the manuscript to briefly discuss potential unintended effects of the proposed RBC modifications on these parameters, providing few references to the in vivo studies in lab animals addressing this key aspect of the development of the RBC-based bioreactors (for example, PMID 27856317).
